# The Microstructural Difference and Its Influence on the Ballistic Impact Behavior of a Near β-Type Ti5.1Al2.5Cr0.5Fe4.5Mo1.1Sn1.8Zr2.9Zn Titanium Alloy

**DOI:** 10.3390/ma13184006

**Published:** 2020-09-10

**Authors:** Xinjie Zhu, Qunbo Fan, Duoduo Wang, Haichao Gong, Hong Yu, Jingjiu Yuan

**Affiliations:** 1School of Materials Science and Engineering, Beijing Institute of Technology, Beijing 100081, China; zhuxinjielyw@163.com (X.Z.); 3120185609@bit.edu.cn (D.W.); 18712780233@163.com (H.G.); yuhong202106@163.com (H.Y.); yuanjj2019@126.com (J.Y.); 2National Key Laboratory of Science and Technology on Materials under Shock and Impact, Beijing Institute of Technology, Beijing 100081, China; 3Beijing Institute of Technology Chongqing Innovation Center, Chongqing 401135, China

**Keywords:** titanium alloy, mechanical properties, microstructure, ballistic impact behavior

## Abstract

In this work, a near β-type Ti5.1Al2.5Cr0.5Fe4.5Mo1.1Sn1.8Zr2.9Zn alloy was hot-rolled at the temperature of 800–880 °C with a thickness reduction of 87.5% and then heat-treated with the strategy of 880 °C/1 h/air cooling (AC) + 650 °C/3 h/AC. The microstructure difference between the hot-rolled and heat-treated titanium alloys and its influence on the ballistic impact behavior of the hot-rolled and heat-treated titanium alloys were analyzed. The microstructural investigation revealed that the average size of the acicular secondary α phase (α_s_) dropped from 75 to 42 nm, and the corresponding amount of this phase increased significantly after heat treatment. In addition, the dislocation density of the α and β phases decreased from 0.3340 × 10^15^/m^2^ and 4.6746 × 10^15^/m^2^ for the hot-rolled titanium alloy plate to 0.2806 × 10^15^/m^2^ and 1.8050 × 10^15^/m^2^ for the heat-treated one, respectively. The high strength of the heat-treated titanium alloy was maintained, owing to the positive contribution of the acicular secondary α phase. Furthermore, the critical fracture strain increased sharply from 19.9% for the hot-rolled titanium alloy plate to 23.1% for the heat-treated one, thereby overcoming (to some extent) the constraint of the strength–ductility trade-off. This is mainly attributed to the fact that the dislocation density and the difference between the dislocation densities of the α and β phases decreased substantially, and deformation localization was effectively suppressed after heat treatment. Damage to the hot-rolled and heat-treated titanium alloy plates after the penetration of a 7.62 mm ordinary steel core projectile at a distance of 100 m was assessed via industrial computer tomography and microstructure observation. The results revealed that a large crack (volume: 2.55 mm^3^) occurred on the rear face and propagated toward the interior of the hot-rolled titanium alloy plate. The crack tip was connected to a long adiabatic shear band with a depth of 3 mm along the thickness direction. However, good integrity of the heat-treated titanium alloy plate was maintained, owing to its excellent deformation capability. Ultimately, the failure mechanism of the hot-rolled and heat-treated titanium alloy plates was revealed by determining the crack-forming reasons in these materials.

## 1. Introduction

In recent years, efforts have focused on reducing the weight of the combat vehicles (e.g., tanks and armored vehicles) with the aim of improving the maneuverability, fuel efficiency, and transportability [1,2,3,4] without reducing the excellent protective capability. Several armor materials have been developed in order to satisfy this weight reduction requirement. Among all materials, titanium alloys exhibited the best combination of high specific strength, low density, and good corrosion resistance [5,6,7]. 

The ballistic impact behavior and corresponding failure mechanism of titanium alloys with different microstructures after heat treatment have been extensively investigated. Zheng et al. [8] investigated the ballistic impact behavior of Ti6Al4V targets against 12.7 mm armor piercing projectile. Targets with various microstructures (lamellar, fully equiaxed, and bimodal) after different heat treatments were investigated. The results revealed that the bimodal microstructure with relatively thick α platelets in the transformed β matrix exhibited better ballistic impact behavior than other microstructures. This was attributed to the regularly spaced propagating features of adiabatic shear bands (ASBs). The observed ballistic impact behavior was consistent with that reported by Lee et al. [9], who studied the ballistic impact behavior of Ti6Al4V alloy with equiaxed and bimodal micristructures after heat treatment against Caliber 50 AP M2. Sun et al. [10,11] investigated the ballistic impact behavior of Ti6Al4V alloys with three microstructures (bimodal, equiaxed, and lamellar) after heat treatment and found that the ballistic impact behavior was closely correlated with ASB intersections. The lowest number of ASB intersections and cracks, and the smallest expanding range of the sideward ASBs occurred in the equiaxed microstructure, which exhibited the best ballistic impact behavior compared with the bimodal and lamellar microstructures. Singh et al. [12] investigated the effect of heat treatment on the ballistic impact behavior of Ti6Al4V alloys against a 7.62 mm deformable projectile. Compared with other plates, the plates were solution-treated below the β-transus temperature and then aged exhibited a higher ballistic impact resistance for higher energy absorption at higher strain rates and greater resistance to adiabatic shear localization. However, a plug formed in each case, which indicated that the same failure mechanism operated in all cases. Li et al. [13] investigated the ballistic performances of a near-β Ti684 alloy under different heat-treatment conditions. The results revealed that the ballistic performance was only partly dependent on the dynamic strength and dynamic hardness. However, for a given case of strength, the performance was improved with an increase of the critical fracture strain. Zheng et al. [14] studied the failure mechanism associated with the ballistic performance of Ti6Al4V targets with equiaxed and lamellar microstructures after heat treatment against a 12.7 mm AP projectile. The failure mechanisms of these targets were ductile hole enlargement and brittle fragmentation, respectively, and the difference between the mechanisms was attributed mainly to the distinctly different propagating features of ASBs. Similarly, Lee et al. [15] studied the ballistic impact behavior of Ti6Al4V plates with equiaxed and bimodal microstructures after heat treatment against a 12.7 mm AP projectile. In both cases, the failure mechanism of the plates was characterized by a ductile growth mode with some radial cracking. However, compared with the equaiaxed microstructure, the bimodal microstructure with higher strength and greater resistance to ASB formation was more resistant to penetration of the projectile.

The above investigation mainly concentrated on the correlation between different types of microstructure and the ballistic impact behavior of the heat-treated titanium alloys. In addition, the effect of microstructural difference between hot-rolling and heat-treatment titanium alloys on the ballistic impact behavior has also been investigated. Burkins et al. [16] studied the ballistic impact behavior of Ti6Al4V plates under different thermomechanical processes. The ballistic impact behavior of these plates against 20 mm fragment-simulating projectiles (FSPs) and 12.7 mm AP projectiles was evaluated. The results revealed that the V_50_ ballistic limit of the plates subjected to either rolling or annealing above β-transus was lower than that of the plates subjected to α+β rolling or annealing. Sukumar et al. [17] investigated the ballistic impact behavior of the rolled and heat-treated β-CEZ titanium alloys against 7.62 mm AP projectiles. In their study, although the rolled plate had higher strength and hardness, the ballistic penetration resistance was only modestly higher than that of the heat-treated plates. 

However, studies considering the effect of the microstructural differences between the hot-rolled and heat-treated titanium alloys on the ballistic performance are relatively rare, and the corresponding influence mechanism is not clear. Therefore, in the present study, microstructures of a near β-type Ti5.1Al2.5Cr0.5Fe4.5Mo1.1Sn1.8Zr2.9Zn alloy [18] after hot-rolling and heat treatment were quantitatively characterized by analyzing the distribution of secondary α phase and calculating the density of dislocations. In addition, the ballistic impact behavior was investigated via industrial computer tomography (CT) characterization of cracks and the microstructural observation of the hot-rolled and heat-treated plates. The failure mechanism associated with ballistic impact behavior was also discussed.

## 2. Materials and Methods

The as-received material in the present study is a near β-type titanium alloy ingot prepared via triple melting in a vacuum arc remelting furnace. The expected composition of the titanium alloy is listed in Table 1, which is the rigorous calculation result based on the compositions of the raw materials before melting in a vacuum arc remelting furnace. The microstructure of the as-cast ingot is shown in Figure 1. It can be seen that the microstructure exhibits a Widmanstatten structure. The β-transus temperature (985 °C ± 5 °C) was determined by means of a metallographic method. After soaking at 970 °C for 1 h, the as-cast ingot with a thickness of 60 mm was rolled to a 7.5 mm plate with a thickness reduction of 87.5%. Subsequently, the hot-rolled titanium alloy plate was divided into two parts, one of which was solution treated at the temperature of 880 °C for 1 h followed by air cooling (AC), which was followed by aging treated at the temperature 650 °C for 3 h followed by air cooling (880 °C/1 h/AC + 650 °C/3 h/AC), hereafter called as the heat-treated titanium alloy plate.

The schematic of the specimen for a quasi-static tensile test is shown in Figure 2a, and the loading direction is parallel to the rolling direction (RD). In addition, the yield strength (YS), ultimate tensile strength (UTS), and percentage elongation of the hot-rolled and heat-treated titanium alloys were measured at the strain rate of 10−3/s using an Instron Universal Testing machine (Instron 5500R, Instron, Boston, MA, USA). The schematic of the specimen for dynamic compression tests is shown in Figure 2b, and the loading direction is parallel to the normal direction (ND). The true dynamic compression stress–strain curves at a high strain rate were calculated from the engineering stress and strain converted from the original pulses of incident, including reflected and transmitted waves collected by the data acquisition system using the Split Hopkinson Pressure Bar (SHPB) system.

Ballistic tests using 7.62 mm ordinary steel core projectile were performed on 300 mm × 300 mm × 7.5 mm hot-rolled and heat-treated titanium alloy plates. The target plates were impacted at a 0° angle of attack, and the projectile was fired with a velocity of $828_{-8}^{+7}$ m/s from a distance of 100 m. To determine the average ballistic behavior, each target plate was subjected to an impact of at least two shots. The distance between any two crater edges was always greater than three times the diameter of the projectile.

Microstructural characterization was performed via optical microscopy (OM) and field emission gun scanning electron microscopy (FEGSEM; Quanta 200FEG, FEI, Hillsboro, OR, USA). The specimens for the OM (ZEISS Axiovert 25, Carl Zeiss AG, Jena, Germany) and SEM observations were mechanically polished after being ground with various (400–2000) grits of SiC paper, and then, they were etched in Kroll’s reagent (10 mL HF, 30 mL HNO3, and 200 mL H2O). Fine details of the microstructure comprising the hot-rolled specimens and heat-treated specimens were characterized by means of transmission electron microscopy (TEM) performed on a FEI Tecnai G2 microscope (FEI, Hillsboro, OR, USA) operated at 200 kV. For TEM observation, the specimens were machined to a diameter of 3.0 mm using an electron discharge method and then polished to 100 μm thickness using various (400–2000) grits of SiC paper. Afterwards, the specimens were prepared via a twin-jet electrochemical polishing method. The dislocation density was calculated by using the electron backscatter diffraction (EBSD) technique combined with the SEM method. The EBSD measurements were performed using a FEI Nova NanoSEM 430 field emission gun scanning electron microscope (FEI, Hillsboro, OR, USA) operated at 20 kV. The specimens for EBSD examination were electropolished (25 °C, 25 V, −30 s, solution components: 6% HClO4, 34% CH3(CH2)3OH, and 60% CH3OH) to eliminate the surface stress and, in turn, achieve the surface quality required for the measurements. 

## 3. Results and Discussion

### 3.1. Microstructures

The microstructure of the hot-rolled titanium alloy is shown in Figure 3. Many grains are quantitatively analyzed using the metallographic analysis software Image Pro Plus v6.0 (Media Cybernetics, Washington, DC, USA). As shown in the figure, the β grains are transformed from coarse β grains into spindle-shaped grains (average aspect ratio: 4). In addition, remnant structures of prior β grain boundaries (GBs) are observed in the microstructure. 

The transformed β phase (β_T_) and lamellae α phase with an average width of 0.5 μm were distributed in the spindle prior β grain (see the upper right-hand corner of Figure 3). Moreover, a considerable amount of acicular secondary α phase (α_s_) with an average size of 75 nm is precipitated in the transformed β phase. The microstructure of the prior GBs in the hot-rolled titanium alloy is revealed by the SEM image shown in the lower left-hand corner of Figure 3. The transformed β phase and equiaxed α phase with an average grain size of 1.3 μm are observed. However, no acicular secondary α phase is observed in the prior GBs.

Figure 4 shows the microstructure of the heat-treated titanium alloy. As shown in the optical micrograph, the microstructure consists of spindle prior β grains and prior GBs, which is similar to that of the hot-rolled titanium alloy. The microstructures inside the spindle prior β grains revealed via SEM (see image in the upper-hand corner of Figure 4) are characterized by a lamellar α phase with an average grain size of 0.9 μm. Additionally, a large amount of acicular secondary α phase with an average width of 42 nm is precipitated in the grains. The microstructure of prior GBs in the heat-treated alloy is revealed in the SEM image shown in the lower left-corner of Figure 4. It can be seen that the microstructure consists of an equiaxed α phase with an average size of 2.2 μm and a significant amount of acicular secondary α phase precipitated in the prior GBs of the heat-treated alloy.

The above quantitative analysis of the microstructures in the hot-rolled and heat-treated titanium alloys reveals some differences, as listed in Table 2. The average sizes of the lamellar α phase and equiaxed α phase in the heat-treated alloy are larger than those of the hot-rolled titanium alloy. However, the average size of the acicular secondary α phase decreases from 75 to 42 nm, whereas the number of the acicular secondary α phase increases significantly after heat treatment, which may have resulted from an increase in the driving force of grain nucleation [19]. In addition, the different microstructures in the GB regions of the hot-rolled and heat-treated titanium alloys as presented in Figure 3 and Figure 4 may be related to the chemical distribution. During the long aging (3 h) of heat treatment, a large number of dislocations at the GB regions provide the nucleation points for the secondary α phase, and sufficient atomic diffusion occurs. In this process, the size of equiaxed α grains increases, and simultaneously, the secondary α grains grow. When two secondary α grains intersect each other, the growth of the secondary α phase stops [20], resulting in a smaller grain size.

A transmission electron micrograph of prior β grains in the hot-rolled and titanium alloys is shown in Figure 5a. It can be seen that the dislocation tangles occur in most regions of the hot-rolled alloy. Furthermore, these tangles, dislocation network, and dislocation cells (see regions indicated by arrows) form in the α/β interface and lead to a substantial increase in the dislocation concentration. However, in contrast to the hot-rolled alloy, dislocation tangles occur in only a few regions in the heat-treated titanium alloy as shown in Figure 5b, indicating that the dislocation density decreases significantly after heat treatment.

The dislocation density (*ρ_GND_*) of the hot-rolled and heat-treated titanium alloys is determined from the statistical data obtained via EBSD measurements. This density is calculated as follows:(1)$$\rho_{GND}=\lambda \frac{KAM}{b\times R}$$
where λ is a constant that depends on the nature of the dislocation’s edge or screw, *KAM* values are obtained from the EBSD software (Channel 5, FEI, Hillsboro, OR, USA), *b* is the magnitude characterizing the Burgers vectors of the dislocations, and *R* is the kernel size taking into account the neighbor order considered in the KAM mapping. The dislocation distribution maps of the hot-rolled and heat-treated titanium alloys are shown in Figure 6, and the corresponding quantitative data are listed in Table 3. The dislocation densities of the α phase and β phase decrease from 0.3340 × 10^15^/m^2^ and 4.6746 × 10^15^/m^2^ for the hot-rolled titanium alloy to 0.2806 × 10^15^/m^2^ and 1.8050 × 10^15^/m^2^ for the heat-treated titanium alloy, respectively. The reduction in the dislocation density of the β phase is considerably greater than that of the α phase. Furthermore, the difference between the dislocation densities of the α and β phases decreases after the heat treatment. This indicates that dislocations in both phases are more homogeneously accommodated than in the hot-rolled state.

### 3.2. Mechanical Properties

The tensile properties of the hot-rolled and heat-treated titanium alloys are determined at room temperature and a strain rate of 10^−3^/s, as shown in Figure 7. In fact, three groups of experiments were carried out on the hot-rolled and heat-treated titanium alloys, respectively, under the same condition, and extremely similar results were obtained, in which only one measurement was selected in this paper. The ultimate tensile strength and the yield strength decrease by 4.8% and 1.7%, respectively, from 1174 MPa and 1066 MPa for the hot-rolled alloy to 1118 MPa and 1048 MPa for the heat-treated alloy. The maximum elongation of the heat-treated alloy (12.7%) is approximately 29.6% higher than that of the hot-rolled alloy (9.8%). The small decrease in strength and the substantial increase in the ductility of the alloy after heat treatment indicates that, compared with the hot-rolled alloy, the heat-treated alloy exhibits a superior combination of strength and ductility. 

The dynamic mechanical properties of the hot-rolled and heat-treated titanium alloys are determined using the SHPB system at room temperature and a strain rate of 3100/s, as shown in Figure 8. Actually, the results represented in Figure 8 for stress–strain measurements are very reproducible under the same conditions, and only one measurement is selected in this paper from three groups of measurements with extremely similar results. The dynamic compression strength of the heat-treated alloy (1780 MPa) is quite similar to that of the hot-rolled alloy (1785 MPa). Moreover, the critical failure strain increases significantly from 19.9% for the hot-rolled alloy to 23.1% for the heat-treated alloy. Significantly, the inset image in Figure 8 shows a greater descent rate after the yield point of the hot-rolled titanium alloy than that of the heat-treated titanium alloy. This indicates that the dynamic softening degree of the hot-rolled titanium alloy is considerably more severe, implying a faster failure process compared to the heat-treated titanium alloy.

The above analysis shows that the mechanical properties of the titanium alloy are effectively improved after the heat treatment. This may be attributed to variations in the dislocation density and the α precipitates. The significant increase in ductility after heat treatment may be attributed to the decrease in the dislocation density (the increase in the density of the α precipitates may (to some extent) be detrimental to the ductility [21,22,23,24]). The dislocation density and the discrepancy between the dislocation densities of the α and β phases decrease substantially after heat treatment. This results in a prolonged deformation process and alleviates the induced inhomogeneous strain, thereby improving the ductility of the titanium alloy. 

The reduction in the strength after heat treatment is postponed effectively for the changes in the α precipitates. Firstly, the sharp increase in the amount of relatively fine acicular secondary α phase in the heat-treated titanium alloy promotes the formation of α phase/β phase interfaces, thereby retarding the strength reduction [25]. Secondly, the segmentation of the acicular secondary α phase relative to the β matrix results in strong constraints on the β matrix. According to the Hall–Petch-like equation $\sigma_{y}=\sigma_{0}+k_{0}/D^{-1/2}$ ($\sigma_{y}$: strength, $\sigma_{0}$: constant that can be considered as either the frictional stress induced by dislocation motion or internal back stresses [26], $k_{0}$: materials constant, and *D*: spacing of the acicular secondary α phase), the decrease trend of the strength after heat treatment is also delayed for the decrease of the *D* value compared with the hot-rolled titanium alloy. 

### 3.3. Ballistic Impact Studies

After the ballistic tests, each impact site is carefully examined. In fact, two impact points were formed on each target plate and exhibited the same ballistic impact behavior, illustrating good reproducibility of the ballistic impact behavior of the hot-rolled and heat-treated titanium alloys. The images of the hot-rolled and heat-treated titanium alloy plates after the ballistic test are shown in Figure 9, which indicates clearly that both plates are not penetrated. However, the dissimilarity of the ballistic performance still exists between the two plates. In the front face of the hot-rolled and heat-treated titanium alloy plates after the tests as shown in Figure 9a,c, the impact crater formed in the hot-rolled plate is narrower and shallower, while that in the heat-treated plate is wider and deeper due to its lower strength. The corresponding rear face of the hot-rolled and heat-treated titanium alloy plates is shown in Figure 9b,d. It can be seen that bulges with cracks are formed on the rear face of each plate. However, compared with the main crack of the heat-treated alloy, the main crack of the hot-rolled plate is considerably wider and longer, owing possibly to the poor ductility of the hot-rolled plate. In addition, in the rear face of the hot-rolled plate, a larger amount of material is involved in the deformation process than that in the heat-treated plate. Similar results were also reported by Singh et al. [27] in the study of comparing the ballistic impact behavior of the hot-rolled high nitrogen steel (HNS) plates and the conventionally used rolled homogeneous armor (RHA) plates. In their study, the HNS plates involved larger strain in the front face and less strain in the rear face compared with the RHA steel plates during the deformation process.

The cracks formed in the hot-rolled and heat-treated titanium alloy plates after ballistic testing are quantitatively characterized via the industrial CT method as shown in Figure 10. The three-dimensional distributions of the main cracks in both titanium alloys as shown in Figure 10a,c reveal that the crack volume of the hot-rolled alloy is considerably larger than that of the heat-treated alloy (2.55 mm^3^ and 0.06 mm^3^ respectively). Magnified views of the main cracks are shown in Figure 10b,d. For the hot-rolled alloy, the maximum crack width (0.22–0.24 mm) occurs in the central region of the bulge. The crack width decreases gradually with the increase of the distance from the central region of bulge, and it varies mainly from 0.10 to 0.17 mm. However, the main crack of the heat-treated titanium alloy undergoes incomplete propagation, and the maximum crack width (0.08 mm) occurs in several separated points. In addition, the crack width ranges mainly from 0.03 to 0.06 mm, which is significantly smaller than that of the hot-rolled alloy. 

For a thorough understanding of the present ballistic results, detailed microstructural studies are performed on the plates after ballistic impact. Using wire-electrode cutting, the specimens after industrial CT investigation are machined along the thickness direction, which crosses the central region of the crack on the surface.

Figure 11 shows optical micrographs of the microstructure comprising the impact crater half section of the hot-rolled titanium alloy plate. The figure indicates that an impact crater with a measured depth of 640 μm emerges in the front face of the plate. In addition, numerous adiabatic shear bands (ASBs) and ASB-induced cracks are distributed on the crater wall. Intersection and bifurcation of the ASBs are observed, indicating that adiabatic shear failure may have assisted the penetration process. In the rear face of the plate, two cracks are nucleated, and these cracks then propagate toward the interior of the plate. In fact, the main crack initiated in the rear face of the plate extends to a large depth of 2.25 mm along the thickness direction. The microstructure of the ASB (region 1 in Figure 11) is shown in Figure 12. It can be seen that a long ASB emanates from the tip of the main crack and expands toward the crater with a vertical depth of 3 mm. The ASBs connected to cracks provide an easy crack propagation path, which accelerates the failure process of the hot-rolled titanium alloy. This phenomenon has also been reported in the studies of other titanium alloys [28,29], steel [30,31], and aluminum alloy [32,33,34] and under the condition of high strain rate.

Figure 13 shows optical micrographs of the microstructure comprising the impact crater half section of the heat-treated titanium alloy plate. The depth of the crater in the heat-treated plate (1400 μm) is considerably greater than that of the hot-rolled plate. In addition to cracks as shown in Figure 13, ASBs occur on the crater wall of the heat-treated plate, indicating that adiabatic shear failure may have assisted the process of projectile target interaction. However, adiabatic shear bands are absent from the rear face of the heat-treated target plate. The only cracks observed in the rear face are relatively fine as shown in Figure 13, which is consistent with the results obtained for industrial CT characterization of the cracks (see Figure 10).

A comparison of the optical micrographs reveals that compared with the hot-rolled titanium alloy plate, the heat-treated titanium alloy plate is more resistant to penetration of the 7.62 mm ordinary steel core projectile. This difference between the ballistic impact behaviors of the plates results mainly from the mechanism of crack formation. That is, the integral area under the dynamic stress–strain curve of the heat-treated titanium alloy is greater than that of the curve associated with the hot-rolled alloy. Therefore, most of the projectile kinetic energy is absorbed by the front face of the plate. The deformation of the rear face is, correspondingly, slighter than that of the front face. Accordingly, no catastrophic crack occurs on the rear face of the heat-treated plate, and thus, the integrity of this plate is better maintained than that of the hot-rolled plate. However, the integral area under the dynamic stress–strain curve of the hot-rolled plate is relatively small and, hence, the kinetic energy of the projectile is inadequately absorbed by the front face of the plate. Therefore, intense deformation on the rear face of the hot-rolled plate is inevitable, and a catastrophic crack forms on this face, leading to a poor ballistic performance of the plate.

## 4. Conclusions

The microstructures of the hot-rolled and heat-treated titanium alloy sheets were quantitatively characterized, and the influence of the microstructures’ difference on the ballistic impact behavior was investigated. The major conclusions of the present study are summarized as follows:After the heat treatment, the average size of the acicular secondary α phase decreased from 75 nm to 42 nm, and the amount of this phase increased substantially. Furthermore, the dislocation density of the α and β phases decreased from 0.3340 × 10^15^/m^2^ and 4.6746 × 10^15^/m^2^ to 0.2806 × 10^15^/m^2^ and 1.8050 × 10^15^/m^2^, respectively.The constraints of strength–ductility trade-off were overcome to some extent after heat treatment. A relatively high strength was maintained after the treatment, owing to the positive contribution of acicular secondary α to the strength. Moreover, the critical fracture strain increased sharply from 19.9% to 23.1% with the substantial reduction in the dislocation density and the difference between the dislocation density of the α and β phases. Consequently, deformation localization was effectively restrained.Industrial CT characterization showed that a catastrophic crack with a maximum volume of 2.55 mm^3^ emerges in the rear face and penetrates the interior of the hot-rolled titanium alloy after the ballistic impact test. The crack tip is connected with a long ASB (depth: 3 mm) along the thickness direction. However, the crack volume of the heat-treated titanium alloy is only 0.06 mm^3^, and the plate maintains better integrity than the hot-rolled plate.The microstructures and mechanical properties of the materials were compared after the ballistic impact test. This comparison revealed that the preferable ballistic impact behavior of the heat-treated titanium alloy plate results from the fact that most of the projectile kinetic energy is absorbed by the front face of the plate. Therefore, the rear face of the plate was effectively protected, in contrast to the rear face of the hot-rolled alloy plate.

## Figures and Tables

**Figure 1 materials-13-04006-f001:**
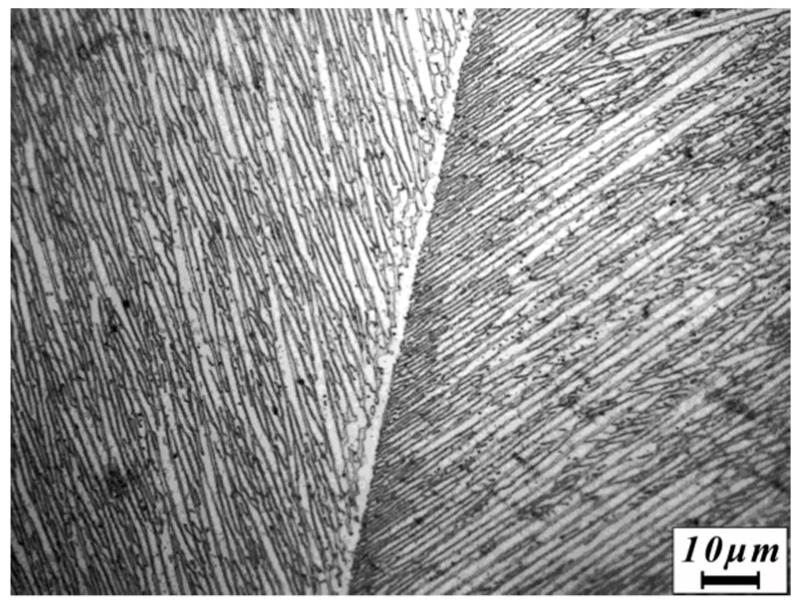
Original microstructure of the as-cast titanium alloy.

**Figure 2 materials-13-04006-f002:**
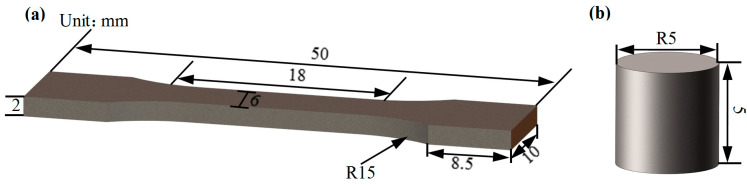
Schematic showing the dimensions of the (**a**) quasi-static tensile specimen and (**b**) dynamic compression specimen.

**Figure 3 materials-13-04006-f003:**
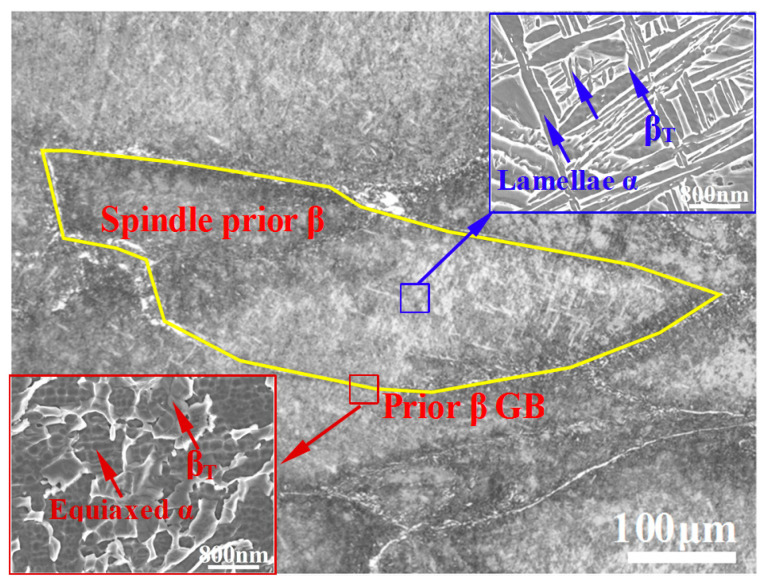
Microstructure of the hot-rolled titanium alloy.

**Figure 4 materials-13-04006-f004:**
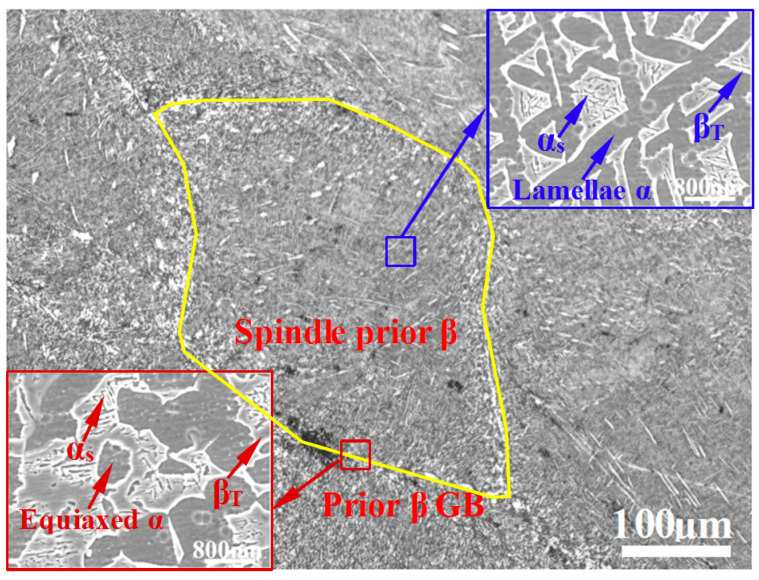
Microstructure of the heat-treated titanium alloy.

**Figure 5 materials-13-04006-f005:**
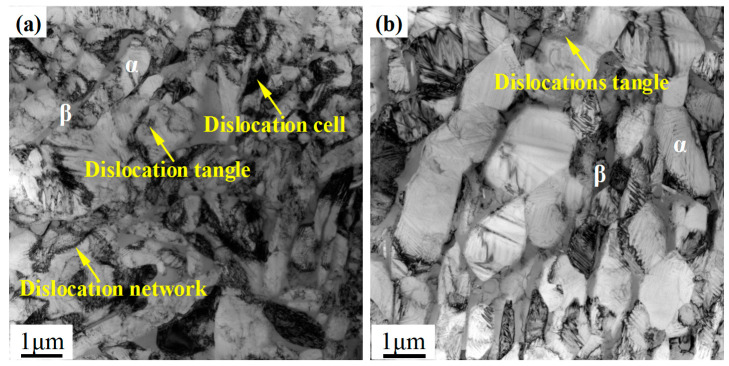
Transmission electron micrographs of the (**a**) hot-rolled specimen and (**b**) heat-treated specimen.

**Figure 6 materials-13-04006-f006:**
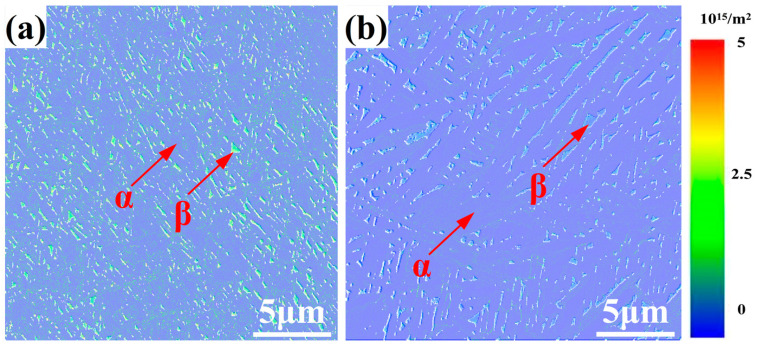
Dislocation distribution maps determined from electron backscatter diffraction (EBSD) measurements of the (**a**) hot-rolled specimen and (**b**) heat-treated specimen.

**Figure 7 materials-13-04006-f007:**
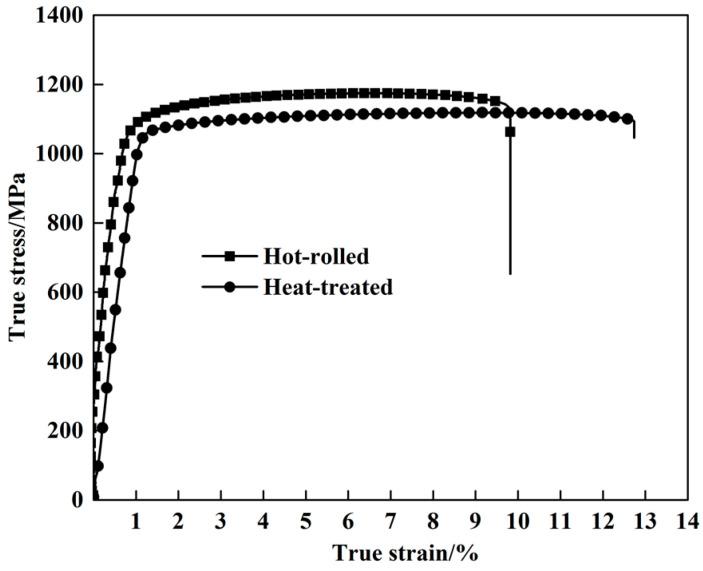
Quasi-static true stress–strain curves of the hot-rolled and heat-treated titanium alloys.

**Figure 8 materials-13-04006-f008:**
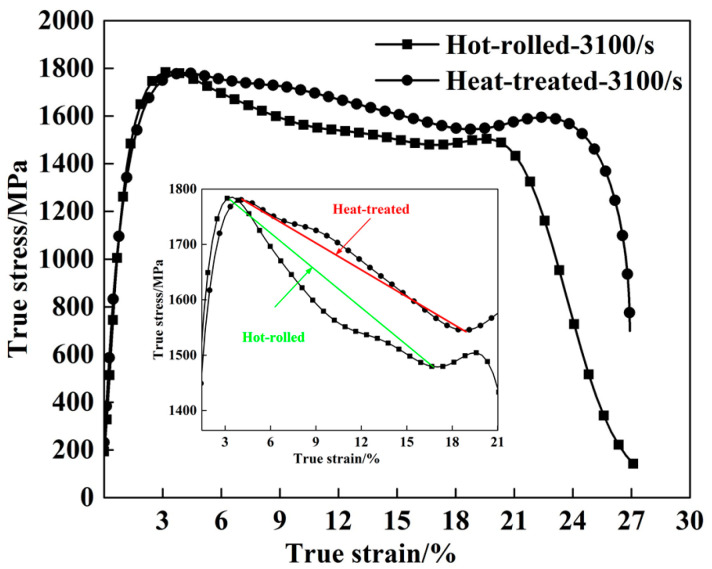
Dynamic true stress–strain curves of the hot-rolled and heat-treated titanium alloys tested at a strain rate of 3100/s. The inset image shows the descent rate after the yield point.

**Figure 9 materials-13-04006-f009:**
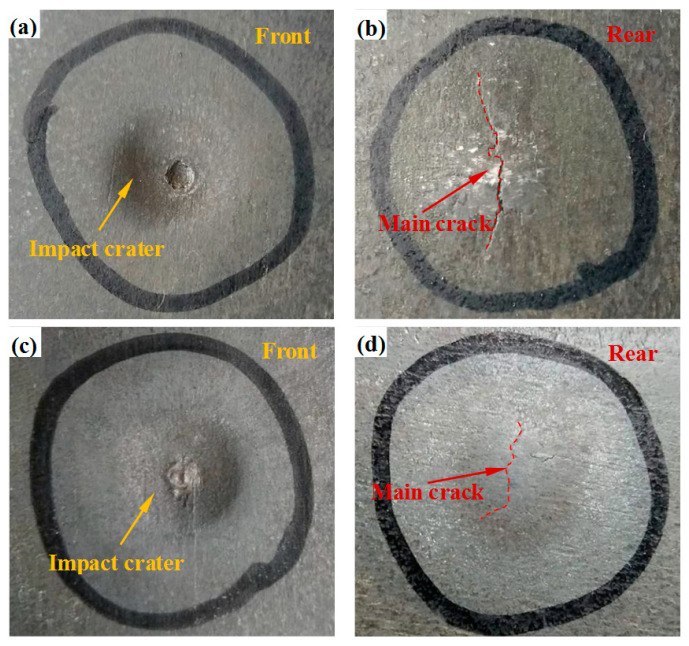
Views of damage patterns on the front face and rear face of target plates after projectile penetration. (**a**) Front view and (**b**) rear view of the hot-rolled titanium alloy plate. (**c**) Front view and (**d**) rear view of the heat-treated titanium alloy plate.

**Figure 10 materials-13-04006-f010:**
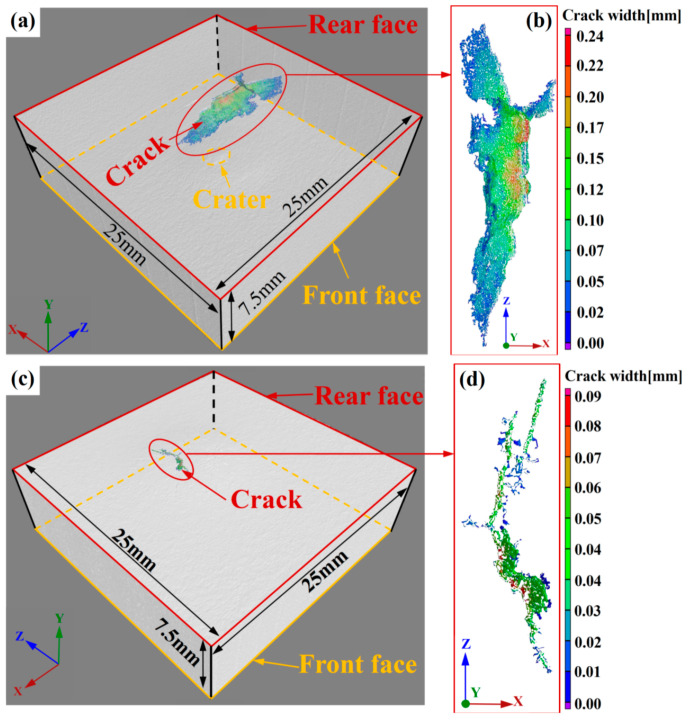
Three-dimensional distribution of cracks, as determined via industrial computer tomography (CT): (**a**) crack distribution in the hot-rolled titanium alloy plate, (**b**) magnified view of the crack in (**a**), (**c**) crack distribution in the heat-treated titanium alloy plate, and (**d**) magnified view of the crack in (**c**).

**Figure 11 materials-13-04006-f011:**
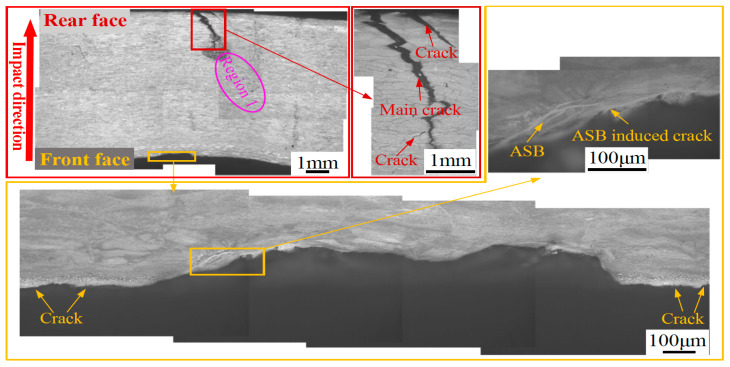
Optical micrographs showing the microstructure of the impact crater half-section of the hot-rolled titanium alloy plate.

**Figure 12 materials-13-04006-f012:**
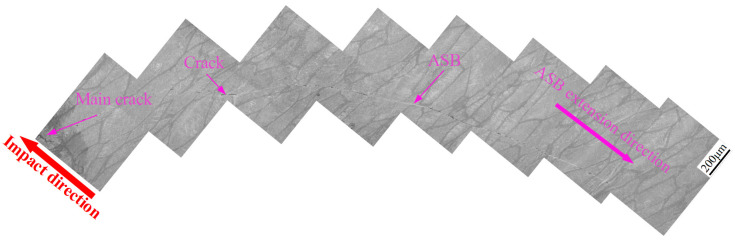
Optical micrographs showing microstructure of region 1 in Figure 11.

**Figure 13 materials-13-04006-f013:**
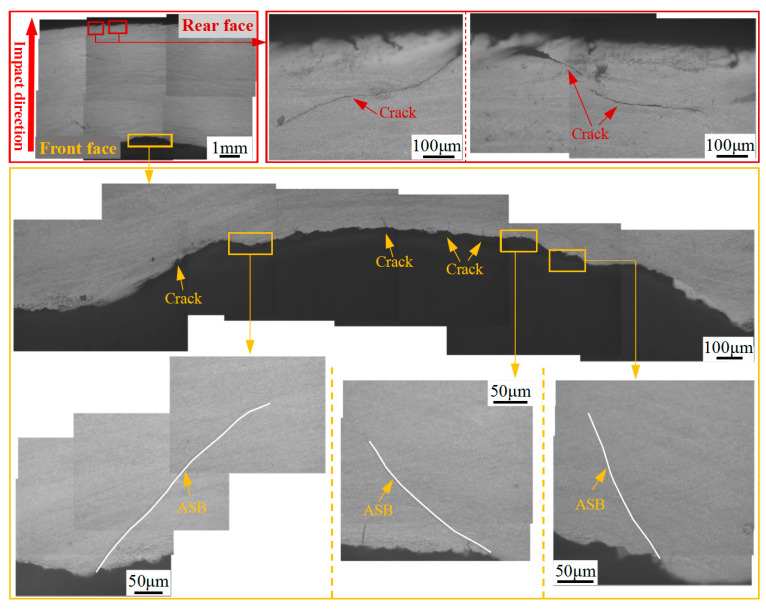
Optical micrographs showing the impact crater half-section of the heat-treated titanium alloy plate.

**Table 1 materials-13-04006-t001:** Composition of the original titanium alloy (wt %).

Element	Al	Cr	Mo	Fe	Zr	Sn	Zn	O	N	H	C	Ti
Wt %	5.12	2.5	4.48	0.52	1.8	1.1	2.9	0.08	0.02	0.002	0.01	Bal.

**Table 2 materials-13-04006-t002:** Microstructural characteristics of the hot-rolled and heat-treated titanium alloys. GB: grain boundary.

Microstructural Characteristics		Hot-Rolled Titanium Alloy			Heat-Treated Titanium Alloy		
		Lamellar α	Equiaxed α	α_s_	Lamellar α	Equiaxed α	α_s_
Grain size	Prior β	0.5 μm	none	75 ± 7 nm	0.9 μm	none	42 ± 5 nm
	Prior β GB	none	1.3 μm	none	none	2.2 μm	42 ± 5 nm
Number of α_s_		Less			More		

**Table 3 materials-13-04006-t003:** Dislocation densities of the hot-rolled and heat-treated titanium alloys.

Dislocation Distribution		Hot-Rolled	Heat-Treated
α/m^2^	<a>	0.2431 × 10^15^	0.0857 × 10^15^
	<c + a>	0.0909 × 10^15^	0.1949 × 10^15^
	Total	0.3340 × 10^15^	0.2806 × 10^15^
β/m^2^		4.6746 × 10^15^	1.8050 × 10^15^

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
