# Peer review of "The Microstructural Difference and Its Influence on the Ballistic Impact Behavior of a Near β-Type Ti5.1Al2.5Cr0.5Fe4.5Mo1.1Sn1.8Zr2.9Zn Titanium Alloy"

_materials, 2020, doi:10.3390/ma13184006_

Round 1
Reviewer 1 Report
The study analyzes the change in ballistic impact behavior of a hot-rolled, near β-type Ti alloy following additional heat-treatment. The material characterization is satisfactory and the results interesting and well presented. There are no major objections to the publication of the paper in its current form. Nevertheless, authors may improve the impact of the study by including discussions on how consistent their observations on the effects of the microstructure and deformed state on adiabatic shear bands formation and impact behavior are with observations in previous ballistic studies of metals (not necessarily limited to Ti alloys).
Minor text editing needed:
- Sentences in lines 49 and 51 use the same wording and express similar concepts, consider removing the sentence in line 49.
- A few typos spotted (e.g. lines 64, 222), revise text for typos.
Reviewer 2 Report
The microstructural difference and its influence on the ballistic impact behavior of a near β-type Ti-5.1Al 3 -2.5Cr-0.5Fe-4.5Mo -1.1Sn -1.8Zr-2.9Zn titanium alloy
This work deals with preparation of hot-rolled and heat-treated titanium alloy sheets and their comprehensive microstructure characterization using various relevant techniques. This research further aims to address ballistic impact behavior of such titanium alloy plates. In overall, the experimental design and applied methods are sufficient to conduct this research. The study aims and the manuscript structure are clear. This research could be of interest to this journal community. In my view some points need to be addressed and discussed for this work:
- How reproducible are results presented in Figure 7 and 8 for stress-strain measurements. Only one measurement is presented in each case for hot-rolled and heat-treated samples.
- How reproducible is crack formation on rear side of a hot-rolled plate. Authors make main emphasis on this observed crack but so far it is based only on one experiment?
- If I understand the data correctly, from Figure 7 and 8 the heat-treated samples were fully destroyed earlier than hot rolled (unless not all data points are plotted for heat treated samples). Could you comment on it?
- The average size of acicular secondary alpha phase decreased from 75 nm to 42 nm. Can you add the error (± ) to the estimated average values?
- Did authors conduct any EBSD measurements after ballistic impact tests to strengthen their conclusions about crack formation?
- Why the nominal composition of the alloy is presented and was not measured before the tests? Where these nominal composition values presented in table 1 were obtained?
Minor comments:
- In introduction, authors define that the main focus in the field is on reducing the weight of the materials. However, later this aspect is not discussed. What is the weight reduction if this alloy would be used?
- Line 133. How was it ensured that the impact angle was 0 degree?
- The surfaces in these study were grinded and not polished, since the polishing nanoparticles were not used.
- Line 165 – prior
- Figure 8. In the figure caption, the inset image should be discussed.
- Sentence 325 should be rephrased, since both of the plates were not penetrated.
Reviewer 3 Report
I have only one comment to this interresting manuscript.
The GB shown in Fig. 1 is evidently a reason of the chemical inhomogeneities resulting in diffrent microstructures in the GB zones of thickness of several tens om μm as presented in Figs. 3 and 4. I would recommned to carry out the additional chemical analysis noncerning the matter and write a short explaining paragraph to it.
Round 2
Reviewer 2 Report
The manuscript was correspondingly improved.
Only a few minor remarks to the authors responses:
“Therefore, the results represented in Figure 7 and 8 for stress-strain measurements are very reproducible. In fact, three groups of experiments were carried out on the hot-rolled and heat-treated titanium alloys, respectively, under the strain rates of 10-3/s and 103/s at room temperature, and the extremely similar results were obtained, in which only one measurement was selected in this paper.”
This reproducibility should be mentioned in the manuscript.
“Moreover, the distance between any two crater edges was always greater than three times the diameter of the projectile. In fact, two impact points formed on each target plate and exhibited the same ballistic impact behavior. Therefore, the crack formation on the rear side of the hot-rolled plate is very reproducible.”
This reproducibility should also be mentioned in the manuscript.
“The material used in the present study is a new high strength and high toughness near β-type titanium alloy developed by Beijing Institute of Technology. In fact, the composition listed in Table 1 is the rigorous calculation result based on the compositions of the raw materials before melting in a vacuum arc remelting furnace. Thus, we deleted the word “Nominal” and revised the caption of Table 1 as: "Table 1 Composition of the original titanium alloy(wt.%)." We really appreciate your kind understanding.”.
As I understood the presented composition is only “expected composition” based on the calculations. It should also be clarified in the manuscript.
Reviewer 3 Report
It is a pity that the authors did not perform a chemical analysis as I recommended. I believe they will do so in the following article.
